# Mapping Gendered Communications, Film, and Media Studies: Seven Author Clusters and Two Discursive Communities

**Kim Britt Pijselman** * 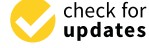 **and Miklós Sükösd**

Department of Communication, Faculty of Humanities, University of Copenhagen, 2300 Copenhagen, Denmark
* Correspondence: kws621@alumni.ku.dk

**Abstract:** This study examined and mapped the extent to which gender became incorporated into the intersecting research fields of communications, film, and media studies. A total of 8054 academic publications from these disciplines, indexed in the Web of Science between 1975 and 2022 ($n_{docs}$ = 8054), were extracted to create two types of bibliometric maps: (a) an author co-citation map, and (b) a co-occurrence map of key terms (taken from keyword lists, titles, and abstracts of publications). Our results revealed a pattern of seven distinct clusters of 995 authors ($n_{authors}$ = 995) in the field. Additional research is needed to analyze the internal structure of these seven clusters, and label them accordingly. The key terms in the same authors' works, however, show a distinctively different pattern, namely a divided, dichotomous, polarized structure ($n_{terms}$ = 720). Judging from this, we hypothesize that gender is discussed in two main ways: either as a critical concept concerning discourses, representations, and other social and cultural constructs, or as a variable in more formal sociological and psychological research designs. The conceptual framework and results of the present study lay the foundation for further research regarding the diverse academic agendas of the seven author clusters, the split nature of their discursive communities, as well as the key difference between the two patterns.

**Keywords:** gender; film; television; radio; communications; film and media studies; communications studies; social sciences; humanities; gender research; bibliometrics; bibliometric research; co-citation; Web of Science; VOSviewer; interdisciplinary



## 1. Introduction

The study of gender has become an important aspect in both the social sciences and the humanities. Diverse conceptualizations as well as a large body of evidence enrich our understanding of the topic of gender in the two academic traditions. At the same time, gender research at the nexus of social sciences and humanities also has important implications for communications, film, and media studies. Communications, film, and media studies itself is an interdisciplinary field that borrows theoretical traditions and empirical methods from both the social sciences and the humanities. For example, studies like Segado-Boj et al. [1] show that television studies can be located in both the social sciences and the arts and humanities disciplines, respectively.

In fact, the study of gender and media as well as film often overlap. The roles of media and film in gender construction and perception have received considerable attention throughout the years. The representation of gender by news media, American and international film and television series, commercial advertising, social media, computer games and other media genres play critical roles in constructing social norms, values, and ideas about social reality and self-perception. Studies have shown the effects that various media and the film/television industry have on gender-related social values, norms, identities, roles, and psycho-social development [2–4]. For example, in terms of personality development, watching stereotypical roles in films and television can significantly impact children's and adolescents' conceptions of gendered self [5,6].

Gill [7] notes that gender-centered approaches in communications, film, and media studies have highlighted the fact that modern societies are stratified along the lines of gender, race/ethnicity, and class. The privileges, disadvantages, and exclusions associated with these dimensions are unevenly distributed. Modern, mediatized societies are increasingly saturated by media, information, and communications technologies [8]. As a result, it is important to study how patterns of inequality, oppression, equality, and empowerment are connected to images, representations, and cultural constructions in and by the media. As media are involved in constructing reality, media are also actively involved in the production of ideas surrounding gender [9]. Furthermore, De Lauretis [10] suggests that gender is the product of various social technologies, such as media and film, but also critical practices and institutional discourses.

It is therefore valuable to look at gender research at the nexus of communications, film, and media studies, not only because these studies conceptualize and conduct research on gender representation issues, but also because the whole field of communications, film, and media studies finds itself between the humanities and the social sciences. There is a growing need to study the gender dimension in communications, film, and media studies more thoroughly, because these two subjects interact in socially and academically important, even crucial ways.

A useful approach to exploring and documenting the state of affairs and trends within a particular academic discipline is to engage in bibliometric research and visualize academic networks. Bibliometric research is a robust quantitative approach to analyzing a large variety of academic networks, ranging from visualizations of co-occurrences of key terms, to citation relations between journals and other publications [11]. For the present research, we conducted analysis by using VOSviewer, a powerful tool to construct and visualize bibliometric networks.

The aim of this paper is to use bibliometric data to explore patterns in the field of communications, film, and media studies and its approach to gender more intensely. More specifically, the objective of the study is to map the intellectual structure of the academic field of communications, film, and media studies from the perspective of how it addresses the topic of gender. Therefore, this paper asked the following four research questions:

1. Which authors contribute to gendered communications, film, and media studies?
2. Which author clusters exist in gendered communications, film, and media studies and how do they relate to each other?
3. Which keywords/terms are most likely to be used in conjunction with gender?
4. Which keyword clusters exist in gendered communications, film, and media studies and how do these relate to each other?

Our study started with a total publication dataset of 8054 documents ($n_{docs}$ = 8054) indexed in the Web of Science database. The first publication was dated in 1975 and the last one in 2022. More specifically, we conducted two types of analysis: (a) a co-citation of authors and (b) a textual occurrence of key terms. The first analysis was conducted to answer the first two research questions, whereas the latter analysis dealt with the last two questions. Our objective was to explore the main intellectual trends in the field, and to identify which theoretical body of works serve as a foundation for the discipline. Results indicate that there are seven distinct clusters of a total of 995 authors ($n_{authors}$ = 995) that interact via main hubs in gendered communications, film, and media studies. At the same time, the application of the term gender in these intersecting fields is clearly divided.

We begin first by clarifying what the term "gender" means. In public discussions and representations, the terms sex and gender are often used interchangeably. Many people take masculinity and femininity as two naturally given categories, rather than seeing gender as socially constructed. In this academic context, the term "sex" is used, on the one hand, to refer to biological features such as chromosomes, hormones, sex organs, and other physical features. On the other hand, the term "gender" is used to refer to the social and cultural meanings that have historically and socially been related to these biological differences [12].

Gender is thus about the social and cultural construction of various notions regarding femininities, masculinities, and other gendered identities, activities, and bodies.

Furthermore, instead of constantly referring to the topic of gender in communications, film, and media studies, the term "gendered communications, film, and media studies" will be used. It should be noted that the term "Communication" in this paper relates to the Web of Science category of communication, whereas the term "communications" refers to the field of communications studies (in relation to media and film studies).

The paper is divided into six sections. This (1) introduction is followed by (2) a brief overview of the literature to locate our study and argue for its relevance. Then, (3) a description of data collection methods and procedures is presented, followed by (4) a descriptive and visual presentations of results, as well as our data analysis. Finally, (5) a discussion and the limitations, as well as (6) the conclusion will be presented.

## 2. Literature

Gendered communications, film and media studies is a field that has yet to be assessed, or even mapped from a bibliometric perspective (see www.vosviewer.com/publications (accessed on 7 September 2022) for a detailed list of bibliographic studies using VOSviewer). Some existing studies in the larger area of humanities are more generalist, such as Leydesdorff et al. [13] who mapped the structure of the Arts and Humanities Citation Index, using techniques that are typically applied in the cases of the Sciences and Social Sciences. Other scholars such as Dharmani et al. [14] employed VOSviewer to map relevant networks and thematically related clusters in academic literature pertaining to the creative industries. In this study, the audiovisual and media industry was examined from organizational, managerial, and industrial perspectives. The topic of gender, however, was not included.

There is only a relatively small body of literature concerned with exploring bibliometric networks within communications, film, and media studies. Most of this research has focused on specific areas in media studies. For example, Leung et al. [15], provided a systematic overview of the academic literature regarding social media. A few other studies take the topic of gender into account, but then utilize VOSviewer to map social media posts and not academic literature. Holmberg and Hellsten [16], for instance, studied gender differences in climate change posts on Twitter, as well as gender differences in the use of affordances on this platform. Their results indicated that even though male and female Twitter users utilized similar language in their tweets, there were clear differences when it came to the use of hashtags and usernames. Women mentioned significantly more campaigns and organizations, whereas men referred more to private people. This study also showed that women focused primarily on the anthropogenic impact on climate change, whereas men usually mentioned usernames with a skeptical stance. A second Holmberg and Hellsten article using VOSviewer examined the response to the fifth Intergovernmental Panel on Climate Change (IPCC) report, but did not account for gender differences [17].

Similarly, in film and television studies, very few bibliometric mapping studies have been carried out. Segado-Boj, Martín-Quevedo and Fernández-Gómez [1] used VOSviewer to map the academic field of television studies. They investigated how this specific field of study came about, what its characteristics are, the streams within it, and the extent to which the rise in publications reflected a consolidated and mature field of research. Researchers have also used VOSviewer to map the language of television scripts in film studies, but not the academic literature of that discipline. Gálvez et al. [18] for instance, analyzed a series of television scripts in the Western film genre, covering almost half a century, and found that male characters were written in a way that expressed a higher level of cognitive abilities than women.

In short, no previous comprehensive study has investigated gendered communications, film, and media studies by using visual bibliometric mapping methods. A critique of the few existing studies could be that their timespans are limited, and their subject definitions and sample sizes too small. Compared to previous studies, our study explores a large

sample size of 8054 publications ($n_{docs}$ = 8054) and covers a considerable timeframe of nearly 50 years, from 1975 till 2022.

## 3. Data Collection: Methods and Procedures

To reiterate, this paper sought out to answer the following four research questions: (1) which authors contribute to gendered communications, film, and media studies; (2) which author clusters exist in gendered communications, film, and media studies and how do they relate to each other; (3) which keywords/terms are most likely to be used in confluence with gender; and (4) which keyword clusters exist in gendered communications, film, and media studies and how do they relate to each other? These research questions were operationalized into specifically designed data collection and analysis questions that could be answered by conducting bibliometric research in the Web of Science database:

1. Which author clusters exist when it comes to the topic of gender in the database?
2. To what extent do the different clusters link (cooperate) with one another in a data visualization map?
3. How is the topic of gender addressed in the two Web of Science categories of Film, Radio, Television; and Communication?
4. What clusters exist when it comes to the topic of gender in these two categories, when we look at co-cited authors and keyword mapping, respectively?

Web of Science is a global academic database that is not affiliated with any publisher. It is a powerful and interdisciplinary research tool. The Web of Science Core Collection (one of the Web of Science's databases) offers a trusted, high-quality collection of journals, books, and conference proceedings in which six different citation indexes are included. For this study, we selected the databases within the Web of Science Core Collection that pertained to the humanities and social sciences. Our study set out to find every possible entry from the start of the databases until the present time, but the first data entry of our keyword ("gender") was found in 1975. Below, we included a list of the databases that were selected, including the timeframe that they covered.

- Social Sciences Citation Index (SSCI)—from 1956 to the present;
- Arts and Humanities Citation Index (AHCI)—from 1975 to the present;
- Conference Proceedings Citation Index—Social Science & Humanities (CPCI-SSH)—from 1990 to the present;
- Book Citation Index—Social Sciences & Humanities (BKCI-SSH)—from 2005 to the present.

Every record in the Web of Science Core Collection also inherits the subject category of its source publication (a journal, book, etc.), also known as its Web of Science Category. A record can be assigned to more than one category. As we addressed the areas of communications, film, and media studies, the two Web of Science categories that were selected were: (i) Communication; and (ii) Film, Radio, Television. The search was further refined by only displaying English entries.

A search for documents pertaining to a specific issue (i.e., gender) can be carried out in two ways: a topic search for words only in the titles; or a topic search for words in the abstract, title, or keyword fields of an article. We tried both and found that the latter option yielded the most relevant records. Since we focus on gendered communications, film, and media studies (implying that gender is a variable that should be taken into consideration as frequently and as much as possible, even if it is not necessarily the most important or sole variable), the word "gender" could be found in either the abstract, title, or keyword fields of an article or book.

A total of 8054 records ($n_{docs}$ = 8054) were downloaded from the Web of Science database by extracting their full records and cited references. Then, data analysis was conducted using VOSviewer, a computer program for constructing maps of bibliometric and other networks. It can be used to construct networks of scientific publications, scientific journals, researchers, research organizations, countries, keywords, or terms. Items in these

networks can be connected by co-authorship, co-occurrence, citation, bibliographic coupling, or co-citation links [11,19]. Even though it is intended primarily for visualizing and analyzing bibliometric networks, it can also be used to visualize social network structures as well as other types of documents.

## 4. Data Analyses and Results

First, this study looked at the information and details regarding the size of the corpus (the total number of relevant articles and books). This included historical trends, countries of publications, academic fields and disciplines, and the most popular publication titles. Then, two analyses were conducted: (a) a network analysis of the bibliographic data, specifically looking at "co-citations" (author map); and (b) a network analysis of the text data, specifically looking at the "co-occurrence" terms (keyword maps). Co-citation can be defined as the frequency with which two documents are cited together [20]. This means that two authors are co-cited if there is a third author that cites both authors together [11]. Co-citation maps can give the overview of the network structure of the academic world [19]. Co-occurrence, on the other hand, happens when two keywords appear together in a document. We present both analyses in tables and network visualizations.

### 4.1. Details of the Corpus

#### 4.1.1. Historical Trends

Across the timespan of 47 years (1975–2022), 8054 articles and books ($n_{docs}$ = 8054) were collected. As Figure 1 shows, the number of academic articles and books addressing gendered communications, film, and media studies has exponentially increased since the 1980s. The purple bars indicate the number of relevant publications throughout the years. The dark-blue line represents the total number of citations of all items in the results set.

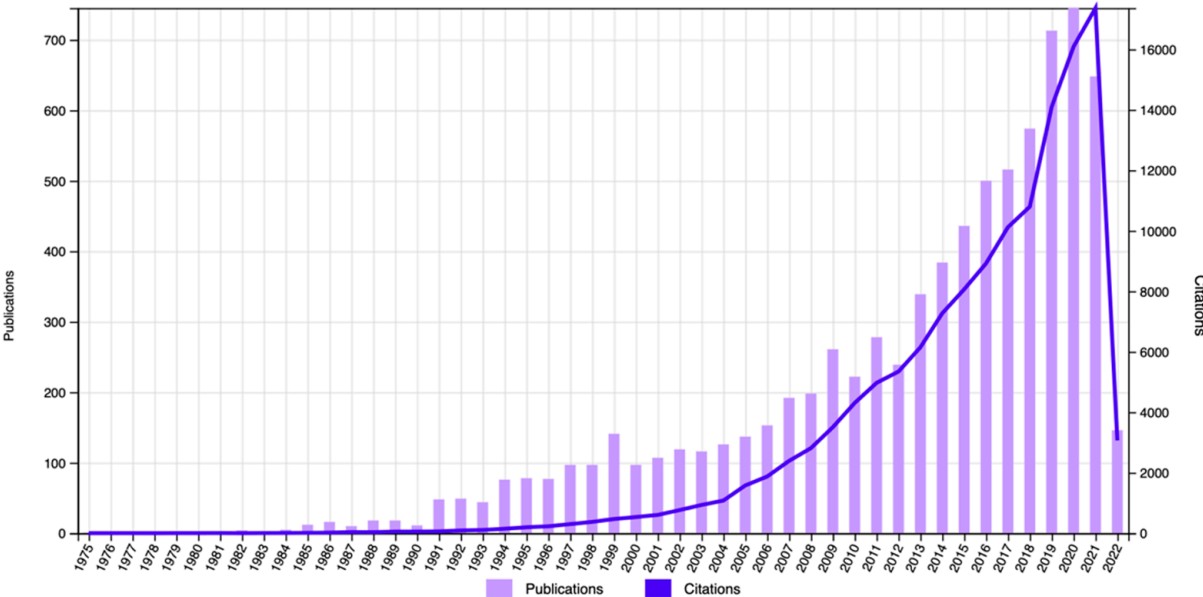

**Figure 1.** Number of publications and citations mentioning the term "gender" in the Web of Science categories of Communication or Film, Radio, Television.

Figure 1 illustrates how popular and prevalent gender became in the communications, film, and media studies literature, especially in recent decades. In fact, we witness a dramatically growing interest in gendered communications, film, and media studies in the early twenty-first century. A significant part of this growth represents the fact that there may also have been newly established communications, film, and media journals that started addressing the topic of gender. For instance, the journal *Feminist Media Studies* was founded in 2001.

It is interesting to note that the number of publications and citations reached an all-time high in the year 2021. On top of that, there is a drop off in data associated with 2022. Since we collected our data on the 5th of April 2022, the data were not yet complete for 2022. Therefore, it is safe to assume that the number of publications and citations changed by the end of the year. It is possible that the number of records in 2022 could have surpassed 2021, and that the exponential grow continues.

### 4.1.2. Countries of Publication

Next, we focus on the country in which the documents were published. In total, entries from 105 different countries were collected. The top ten countries are displayed below in Table 1.

**Table 1.** Top 10 publishing countries.

| Web of Science Country Category | Number of Entries |
| --- | --- |
| USA | 4063 |
| England | 823 |
| Australia | 470 |
| Canada | 366 |
| People's Republic of China | 228 |
| The Netherlands | 222 |
| Spain | 221 |
| Israel | 151 |
| Belgium | 119 |
| New Zealand | 99 |

One notices a clear dominance of American journals and book publishers in gendered communications, film, and media. Furthermore, the top four are all English-speaking countries, including England, Australia, and Canada. This is not surprising, considering that the filter on the dataset selected only documents written in English. However, this is also in line with previous research that showed the domination of Anglophone journals and book publishers in international academic publications [21]. Previous research about the creative industries also indicated that the United Kingdom and the United States dominated that area [14]. It is also noteworthy that there are no African or Latin American countries in the top ten list, and Asia is represented only by the People's Republic of China.

### 4.1.3. Academic Fields and Disciplines

We selected two Web of Science categories in the search query: Communication and Film, Radio, Television, respectively. However, the results indicated that many journal articles and books in gendered communications, film, and media studies were interdisciplinary and listed in other Web of Science categories. In total, relevant academic publications in 79 adjacent fields were found. The top ten categories are displayed in Table 2.

**Table 2.** Top 10 Web of Science categories.

| Web of Science Category | Number of Entries |
| --- | --- |
| Communication | 7066 |
| Film, Radio, Television | 1474 |
| Psychology Social | 899 |
| Women's Studies | 798 |
| Family Studies | 746 |
| Sociology | 504 |
| Cultural Studies | 440 |
| Linguistics | 425 |
| Political Science | 352 |
| Business | 311 |

As Table 2 indicates, the journals and books that publish on gendered communications, film, and media studies do not just belong to the Communication and Film, Radio, Television categories. As mentioned, both gender studies and communications, film, and media studies constitute interdisciplinary areas that find themselves at the nexus of the social sciences and humanities. Most items in Table 2 cannot be classified into only one of the two large schools of thought. One could argue that Cultural Studies and Linguistics largely belong to the humanities approach, whereas Psychology Social, Women's Studies, Sociology, Political Science, and Business traditionally have more affinity with social sciences. However, all these fields have interdisciplinary tendencies.

In fact, when looking at the self-description of journals and publishers in these areas, it becomes clear that most of them follow cross- and/or interdisciplinary academic agendas, have diverse theoretical programs, and allow for methodological pluralism, which means that they include quantitative, qualitative, and mixed methodologies. Thus, the findings in the table above also signal that many journals and book publishers that engage in gendered communications, film, and media studies are cross-disciplinary, and feature diverse methodological orientations.

What remains interesting about the data in Table 2 is that the Communication category is much larger than all other categories combined. A possible explanation could be that by selecting the Communication category, not only "communications" in the sense of media communication studies were included, but the entire field of communication studies, including communication in social and (intra-)personal relations, the art of rhetoric and persuasion, or even organizational communications in the health or other sectors. For instance, a study by Tavares et al. [22] found that by looking at gender in the communication discipline, articles were included that pertained to journalism, film, image, and audiovisualities, media studies, advertising and marketing, and public relations and organizational communications.

### 4.1.4. Journals Publishing Gendered Communications, Film, and Media Studies

In total, 1102 titles of journals and book series were found that published gendered communications, film, and media studies. The top fifteen journals, including their earliest publication date, are displayed in Table 3.

**Table 3.** Top 15 Publication titles.

| Publication Title (Incl. Earliest Publication Date) | Number of Articles |
| --- | --- |
| *Journal of Social and Personal Relationships* (1984) | 471 |
| *Feminist Media Studies* (2001) | 462 |
| *Personal Relationships* (1994) | 254 |
| *New Media & Society* (1999) | 212 |
| *International Journal of Communication* (2007) | 151 |
| *Discourse & Society* (1990) | 148 |
| *Information, Communication & Society* (1998) | 147 |
| *Journal of Broadcasting and Electronic Media* (1957) | 137 |
| *Journal of Health Communication* (1996) | 127 |
| *Media, Culture & Society* (1979) | 124 |
| *Journal of Language and Social Psychology* (1982) | 124 |
| *Journal of Communication* (1951) | 115 |
| *Continuum: Journal of Media Cultural Studies* (1988) | 111 |
| *Journalism & Mass Communication Quarterly* (1924) | 110 |

Table 3 above supports the earlier argument that a firm distinction between the social sciences and humanities can become blurred. When it comes to the journals that publish gendered communications, film, and media studies, most journals explicitly describe themselves as either interdisciplinary, transdisciplinary, multidisciplinary, cross-disciplinary, or crossing disciplinary and sub-field boundaries. Only the *Journal of Health Communication* places itself in the social sciences category [23].

*4.2. Network Analysis Based on Co-Citation*

4.2.1. Most Cited Authors

We also aim at finding out how clusters of authors were formed in gendered communications, film, and media studies, and to which extent these clusters cooperated with one another. First, we looked at the most prominent authors in the field, by taking individual citations and co-citations as units of analysis. Based on this bibliographic data, a map was created using co-citation as a type of analysis. Co-citation can be defined as the frequency with which at least two authors are cited together by other authors in the field. If at least one other author cites two authors in common, the cited authors are said to be co-cited [20]. By measuring co-citation, the relatedness of authors is determined based on the number of times they are cited together. In this way, clusters of authors could be explored and visually represented.

Co-cited authors were calculated by using "full counting", meaning that each author receives one full count every time they were co-cited, even if they had co-authored the cited paper with someone else. The citation threshold was 20, meaning that the minimum number of citations that each author needed to be included in the visual map was 20. Of the 142,860 authors that were extracted, 2302 met this threshold. For each of these 2302 authors, the total strength of the co-citation links with other authors was calculated. In total, the top 1000 authors with the greatest number of co-citations were selected for analysis. A filtering procedure removed five organizations that had authored reports (i.e., the PEW Research Center, Global Media Monitoring Project, World Economic Forum, World Bank, and Entertainment Software Association). In the end, a working set of 995 scholars ($n_{authors}$ = 995) was chosen for this study.

In Table 4, the top fifty co-cited authors are ranked according to their respective results regarding citations and co-citations. The "Citations" column indicates the number of citations made to a cited author, which indicates the total amount of times the author's name was cited with regard to gendered communication, film, and media studies. The "Total Link Strength" column, on the other hand, refers to the total link strength of a particular author, which reflects the authors' scientific impact in the academic community. The colors in Table 4 refer to the author's respective cluster, which will be shown later in Figure 2.

**Table 4.** Top 50 authors based on co-citations.

| Author | Cluster | Citations | Total Link Strength |
|---|---|---|---|
| Butler, J. | Red | 928 | 13,535 |
| Gill, R. | Red | 660 | 10,426 |
| Foucault, M. | Red | 620 | 10,201 |
| Bandura, A. | Green | 548 | 9552 |
| Goffman, E. | Purple | 525 | 9105 |
| Hall, S. | Red | 464 | 8887 |
| McRobbie, A. | Red | 457 | 8098 |
| Eagly, A.H. | Dark blue | 454 | 7901 |
| Bourdieu, P. | Red | 468 | 7533 |
| Connell, R. | Red | 442 | 6558 |
| Cohen, J. | Dark blue | 363 | 5807 |
| Tannen, D. | Purple | 359 | 6285 |
| Livingstone, S. | Yellow | 348 | 5902 |
| Zillmann, D. | Green | 327 | 14,325 |
| Hargittai, E. | Yellow | 324 | 6981 |
| Herring, S.C. | Yellow | 310 | 5298 |

**Table 4.** *Cont.*

| Author | Cluster | Citations | Total Link Strength |
|---|---|---|---|
| Mulvey, L. | Red | 304 | 4689 |
| Buss, D.M. | Dark blue | 293 | 4343 |
| Billings, A.C. | Turquoise | 275 | 4574 |
| Gerbner, G. | Green | 263 | 5028 |
| Sprecher, S. | Dark blue | 252 | 4591 |
| Bem, S.L. | Green | 251 | 4921 |
| Kenny, D.A. | Dark blue | 248 | 3957 |
| Walther, J.B. | Yellow | 244 | 4605 |
| Burgoon, J.K. | Dark blue | 242 | 4817 |
| Fairclough, N. | Purple | 241 | 4356 |
| Van Zoonen, L. | Turquoise | 238 | 4153 |
| Mulac, A. | Purple | 236 | 4344 |
| Ross, K. | Turquoise | 233 | 4222 |
| Burlescon, B.R. | Dark blue | 226 | 5259 |
| Cameron, D. | Purple | 221 | 4171 |
| Signorielli, N. | Green | 221 | 3659 |
| Hooks, B. | Red | 211 | 3118 |
| Anderson, C.A. | Green | 207 | 3990 |
| Hardin, M. | Turquoise | 207 | 2920 |
| Eckert, P. | Purple | 206 | 3359 |
| Holmes, J. | Purple | 203 | 4485 |
| Kahn, K.F. | Turquoise | 203 | 3468 |
| Hall, J.A. | Dark blue | 200 | 2979 |
| Rubin, A.M. | Green | 199 | 8191 |
| Ward, L.M. | Green | 199 | 4117 |
| Gottman, J.M. | Dark blue | 199 | 3770 |
| Boyd, D. | Red | 196 | 5269 |
| Rusbult, C.E. | Dark blue | 192 | 3298 |
| Tajfel, H. | Green | 188 | 4191 |
| Tuchman, G. | Turquoise | 188 | 3908 |
| West, C. | Purple | 188 | 3191 |
| Collins, P.H. | Red | 182 | 2292 |
| Van Deursen, A. | Yellow | 180 | 3352 |
| Mikulincer, M. | Dark blue | 179 | 3538 |

Regarding a technical detail, we discovered that some authors in our top fifty had two separate nodes in the network since they were initially quoted with only their first initial (e.g., J. Butler), but later also with their full name (e.g., Judith Butler). For authors with double nodes, their relevant citations and total link strengths were combined. This procedure was carried out for the following authors: Butler, Foucault, Goffman, Knobloch-Westerwick, Hall, McRobbie, Hargittai, Bourdieu, Connell, Tannen, and Mulvey.

It is apparent from this table that gender theorist Judith Butler is the most prominent cited scholar in the field of gendered communications, film, and media studies. This is understandable, since her influential essay *Performative Acts and Gender Constitution: An Essay in Phenomenology and Feminist Theory* became a cornerstone of feminist theory

by making a distinction between the terms "gender" and "sex", and highlighting the performative acts of gender [24].

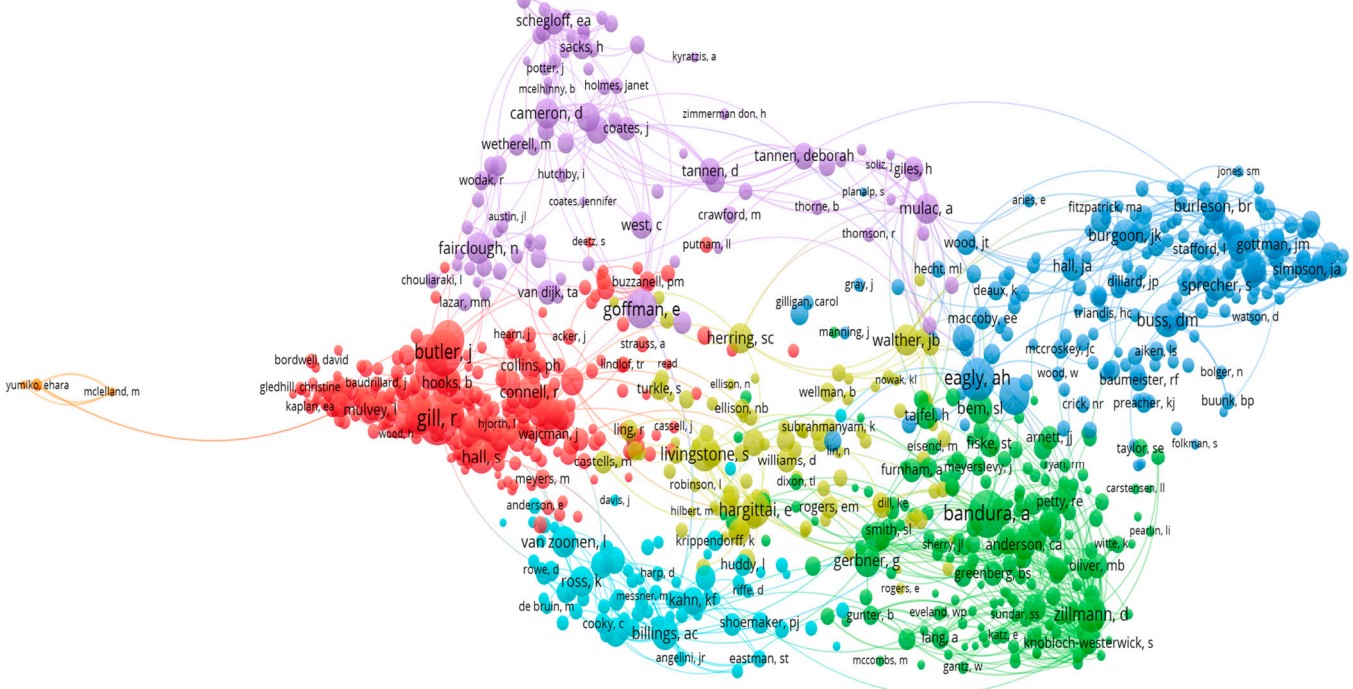

**Figure 2.** Network visualization of co-citation with weights based on co-citation.

The largest group of authors in the list could be considered theoreticians of gender and culture, meaning that they engage in conceptualizations of gender as a social–cultural construct (see Table 4). Authors in this group include Judith Butler, Rosalind Gill, Michel Foucault, Angela McRobbie, and Stuart Hall. They belong to the red cluster. Tied in first place are also the authors from the dark-blue cluster, which includes world-renowned psychologists such as Alice Eagly, Jonathan Cohen, and David Buss. The third cluster with the highest number of authors in the top fifty is the green cluster, which features scholars that focus on media psychology and communication research such as Albert Bandura, Dolf Zillmann, and George Gerbner. Subsequently, the purple cluster with Erving Goffman, Deborah Tannen, and Norman Fairclough can be described as scholars focusing on sociology, linguistics, and discourse analysis. The turquoise cluster focuses on news media, journalism, popular culture, and gender with scholars such as Andrew C. Billings, Liesbet Van Zoonen, and Karen Ross. The yellow cluster includes social media and communications research, including Sonia Livingstone, Eszter Hargittai, and Susan C. Herring. Not one author from the orange cluster is present in the top fifty, which is a small cluster of predominantly Japanese communications, film, and media scholars. It is interesting to note that the orange cluster is the sole one that is organized according to a specific country. Other clusters feature scholars from various countries.

Table 5 provides an overview of seven groups of the top fifty authors based on co-citation. These seven groups are identical to the seven clusters of authors that we identified with network analysis in the field of gendered communications, media, and film studies. To make the discussion easier, we identify the seven clusters with different colors. In the next section, the clusters are explained in more detail. For the visual representations of the seven clusters, see Figure 2.

**Table 5.** Overview of distribution in seven clusters of top 50 authors based on co-citations.

| Cluster | Number or Authors in Top 50 | Total Number of Citations |
|---|---|---|
| Red | 11 | 4932 |
| Dark blue | 11 | 2848 |
| Green | 9 | 2403 |
| Purple | 8 | 2179 |
| Turquoise | 6 | 1344 |
| Yellow | 5 | 1406 |
| Orange | 0 | 0 |

4.2.2. Seven Clusters of Authors Based on Co-Citation

This section explores how gendered communications, film, and media authors are spatially located in a visualization of their bibliometric network. Figure 2 presents a network visualization of co-citations. It is essentially a visualization of Table 4 but instead of the top fifty, it includes all 995 authors ($n_{authors}$ = 995) in the discipline.

In Figure 2, each author is represented by a circle. The size of the author's circle is determined by the number of co-citations an author has. This implies that the larger an author's circle is, the more times that author has been co-selected with other authors when they discuss gendered communications, film, and media studies.

The color of an author is determined by the cluster to which the author belongs. Lines between authors represent co-citations, meaning that two authors are cited together by a third author. By default, at most 1000 links are visually displayed, representing the 1000 strongest links between all authors. However, all links were taken into consideration when creating the network maps and calculating the total link strength.

The distance between two authors in the visualization approximately indicates the relatedness of the author in terms of co-citation links. In other words, the closer two authors are located to each other, the stronger their relatedness. This means that when authors are closer to each other, they are more likely to be co-cited and, thus, have topical or some other substantial similarity.

Figure 2 shows an overview of the seven distinct clusters that represent a total of 995 authors ($n_{authors}$ = 995). The red cluster has the highest number of scholars with 311 authors. The most cited authors in the cluster are Butler, Gill, Foucault, Goffman, Hall, and McRobbie. It is a theoretically oriented and multidisciplinary cluster, involving gender theory, cultural and social theory, sociology, social psychology, and feminism.

The second most populous cluster is the green one with 211 authors. The most cited authors in this cluster are Bandura, Zillmann, and Gerbner. It is followed by the dark-blue cluster (192 authors), the yellow cluster (99 authors), the purple cluster (95 authors), the turquoise cluster (83 authors), and, finally, the orange cluster, with only 4 authors.

It is apparent from Figure 2 that the strongest clusters are the red, green, and dark blue ones, respectively. However, the green and dark-blue clusters are more closely connected with one another than the red cluster is with either one of them. Generally, the path length between a green and a blue node is quite short. A red node, on the other hand, needs to go through several other clusters in order to become connected to a green or dark-blue node. There are only a few central nodes that function as connection hubs that link the biggest clusters to each other.

Overall, the green cluster can be considered to have the strongest position in the bibliometric network of gendered communications, film, and media studies. Green authors have strong connections to other authors in the network, whether this is within their own green cluster or with other clusters. To reach the most influential clusters (either the red or dark-blue cluster), authors must cross through the green cluster. This makes the authors in the green cluster true interdisciplinary scholars who bridge the gap between the different traditions in the field.

*4.3. Network Analysis Based on Text Data*

4.3.1. Most Used Keywords

This section aims at answering how the topic of gender is addressed in the respective Web of Science categories. It was of great interest to discover which keywords/terms are used in publications within the field of gendered communication, film, and media studies, since they reflect the core content of a publications. Therefore, a co-occurrence network map of keywords/terms, based on text data, was created. The same Web of Science bibliographic database set was used that previously visualized the author network. The keywords/terms were extracted from the title and abstract fields of the same academic publications.

As for technical procedures, this study adopted a binary counting method. This implies that only the presence or absence of a term in a document mattered. The amount of occurrences of a term in one document was not considered. The minimum threshold for the number of times a word had to occur in the database was 20 times. Of the 91,704 terms that were extracted from the database, 1248 met this threshold. From each of the 1248 terms, a relevance score was calculated using VOSviewer, which offers a system that evaluates terms based on their substance and subject relevance. Based on this score, the top 60 percent of the most relevant terms were selected. Therefore, the final number of terms that were taken into consideration was 749.

Then, a manual filtering procedure removed the terms that had nothing to do with gendered communications, film, and media studies. These were predominantly general words that are often used in standard academic writing, but they had nothing to do with the specific areas and would not improve our understanding of the various approaches within the field. The words and word co-occurrences that were excluded from the analyses were the following: "article"; "design methodology approach"; "previous research"; "current study"; "i.e.",; "month"; "week"; "mage"; "piece"; "someone"; "year old"; "past research"; "present research"; "results highlight"; "March"; "liking"; "important implication"; "future study"; "December"; "past decade"; "prior research"; "January"; "prior research"; "ease"; "research limitations implication"; "research highlights"; "selfy"; "future direction"; "particular attention"; "move"; "past"; "today"; and "none". At the end of this cleansing process, a working set of 720 keywords ($n_{term}$ = 720) was identified.

In Table 6, the top fifty keywords are ordered according to their frequency of occurrence. The "Occurrences" attribute indicates the total number of documents in which a keyword was found. The colors of the terms correspond to their assigned clusters in Figure 3, which shows that there are two major clusters: red and green. It is important to note that these two colors do not correspond to the colors in the previous tables and figures.

**Table 6.** Top 50 keywords according to occurrences.

| Term | Cluster | Occurrences | Relevance |
|---|---|---|---|
| Relationship | Green | 1219 | 0.32 |
| Effect | Green | 1020 | 0.69 |
| Age | Green | 781 | 0.55 |
| Discourse | Red | 769 | 1.28 |
| Participant | Green | 769 | 0.54 |
| Implication | Green | 732 | 0.33 |
| Culture | Red | 700 | 0.65 |
| Identity | Red | 682 | 0.50 |
| Model | Green | 683 | 0.46 |
| Behavior | Green | 682 | 0.80 |
| Representation | Red | 629 | 1.04 |
| Perception | Green | 599 | 0.46 |

**Table 6.** *Cont.*

| Term | Cluster | Occurrences | Relevance |
|---|---|---|---|
| Factor | Green | 587 | 0.62 |
| Race | Red | 566 | 0.50 |
| Level | Green | 552 | 0.50 |
| Type | Green | 534 | 0.37 |
| Survey | Green | 519 | 0.81 |
| Individual | Green | 512 | 0.48 |
| Sample | Green | 511 | 0.97 |
| Male | Green | 483 | 0.39 |
| Attitude | Green | 477 | 0.46 |
| Film | Red | 465 | 1.56 |
| Information | Green | 453 | 0.45 |
| Politic | Red | 436 | 1.41 |
| Female | Green | 434 | 0.50 |
| Space | Green | 420 | 1.08 |
| Narrative | Red | 416 | 1.13 |
| Sex | Green | 410 | 0.31 |
| Child | Green | 402 | 0.33 |
| Association | Green | 391 | 1.16 |
| Gender difference | Green | 385 | 0.90 |
| Image | Red | 382 | 0.65 |
| Power | Red | 382 | 0.59 |
| Sexuality | Red | 364 | 0.87 |
| Support | Green | 362 | 0.51 |
| Audience | Red | 360 | 0.50 |
| Television | Red | 357 | 0.43 |
| Partner | Green | 347 | 1.83 |
| Body | Red | 331 | 0.68 |
| World | Red | 329 | 0.53 |
| Characteristic | Green | 329 | 0.47 |
| Text | Red | 328 | 1.09 |
| Education | Green | 325 | 0.55 |
| Character | Red | 322 | 0.55 |
| Class | Red | 321 | 0.84 |
| Internet | Green | 321 | 0.61 |
| Story | Red | 308 | 0.79 |
| Outcome | Green | 297 | 0.86 |
| Performance | Red | 291 | 0.41 |
| Student | Green | 290 | 0.81 |

The table above illustrates that the words with the highest occurrence mostly belong to the green cluster: 30 out of the 50 words belong to that respective cluster. From the list of these fifty keywords, we can infer that the term gender is not only used in a theoretical and social constructivist context (the way this paper initially defined the word gender in the introduction), but also in statistical, quantitative, and categorical contexts. This table also shows that the word gender is often used in the same contexts as "sex", even though they refer to two separate concepts. It is also interesting to note that the term "male" is

associated in more publications with gender than "female" is. Furthermore, the list features predominantly social sciences terminology which can be explained by the dominance of the discipline of communication studies, as previously noted in our sample discussion (see Table 2).

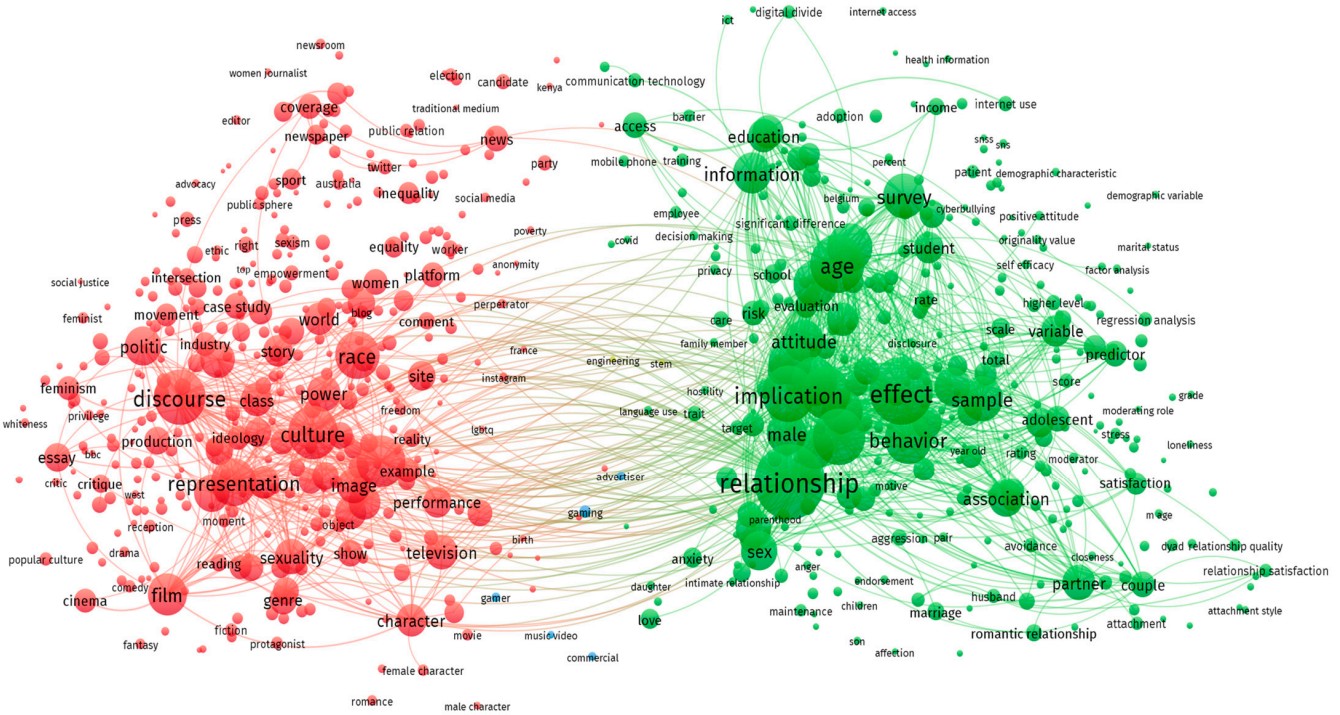

**Figure 3.** Network visualization of term co-occurrence map based on text data with weights based on occurrences.

Table 7 below shows that the green cluster also dominates when it comes to the total amount of occurrences. When all the green keywords are added together, their occurrences almost double the red keywords from the top fifty. Again, an explanation for this is the dominance of the communications-related publications in the sample (see Table 2).

**Table 7.** Overview cluster distribution of top 50 keywords according to occurrences.

| Cluster | Number of Entries in Top 50 | Total Occurrences |
|---|---|---|
| Green | 30 | 15,826 |
| Red | 20 | 8738 |

Below, the top fifty terms are ordered according to their relevance. Again, the colors of the terms correspond to their assigned cluster colors in Figure 3.

Table 8 above shows the terms that are most relevant when it comes to gendered communications, film, and media studies. The data shows that the list of words completely changes when they are ordered according to relevance. Terms with a high relevance score tend to represent topics covered by the gender-specific text data, while terms with a low relevance score tend to be of a general nature and tend not to be representative of any specific topic. Not one word from Table 6 is included in this new list. Again, the majority of the words belong to the green cluster, whilst the red cluster remains underrepresented. This means that we can infer that the words in the green cluster hold more relevance when it comes to gendered communications, film, and media studies compared to the red cluster. Table 8 also indicates that most words refer to research conducted in the fields of communication and inter-personal relationship studies.

**Table 8.** Top 50 keywords according to relevance.

| Term | Cluster | Occurrences | Relevance |
| --- | --- | --- | --- |
| Marital satisfaction | Green | 31 | 3.92 |
| Partner effect | Green | 26 | 3.75 |
| Attachment style | Green | 34 | 3.52 |
| Married couple | Green | 33 | 3.49 |
| Actor partner independence model | Green | 28 | 3.40 |
| Relationship satisfaction | Green | 80 | 3.20 |
| Depressive symptom | Green | 39 | 2.91 |
| Relationship quality | Green | 63 | 2.76 |
| Post-feminism | Red | 42 | 2.67 |
| Spouse | Green | 76 | 2.56 |
| Whiteness | Red | 39 | 2.51 |
| Aesthetic | Red | 48 | 2.49 |
| Couple | Green | 222 | 2.45 |
| Cinema | Red | 189 | 2.43 |
| Post-feminism | Red | 28 | 2.42 |
| Heterosexual couple | Green | 32 | 2.38 |
| Infidelity | Green | 26 | 2.37 |
| Moderating effect | Green | 35 | 2.35 |
| Romantic partner | Green | 62 | 2.32 |
| Self-report | Green | 35 | 2.32 |
| Adulthood | Green | 43 | 2.31 |
| Close relationship | Green | 50 | 2.29 |
| Dyad | Green | 64 | 2.29 |
| Emotional support | Green | 31 | 2.26 |
| Social justice | Red | 26 | 2.25 |
| Attachment | Green | 79 | 2.20 |
| E-mail | Green | 23 | 2.17 |
| Critic | Red | 43 | 2.17 |
| Grade | Green | 36 | 2.16 |
| Feminism | Red | 220 | 2.16 |
| Feminist | Red | 68 | 2.14 |
| Moderating role | Green | 26 | 2.13 |
| Popular culture | Red | 72 | 2.12 |
| National identity | Red | 35 | 2.12 |
| Closeness | Green | 44 | 2.11 |
| Trope | Red | 73 | 2.09 |
| Gender politic | Red | 71 | 2.09 |
| Hegemony | Red | 51 | 2.06 |
| Mediating role | Green | 22 | 2.06 |
| Indirect effect | Green | 35 | 2.05 |
| Female body | Red | 31 | 2.04 |

**Table 8.** *Cont.*

| Term | Cluster | Occurrences | Relevance |
|---|---|---|---|
| Hollywood | Red | 37 | 2.03 |
| Factor analysis | Green | 40 | 2.02 |
| Satisfaction | Green | 175 | 2.00 |
| Neoliberalism | Red | 33 | 1.98 |
| Internet access | Green | 33 | 1.97 |
| Loneliness | Green | 34 | 1.97 |
| Novel | Red | 55 | 1.97 |
| Adolescence | Green | 50 | 1.96 |
| Essay | Red | 244 | 1.95 |

Table 9 below illustrates that the green cluster again dominates when it comes to the highest relevance scores.

**Table 9.** Overview cluster distribution of top 50 keywords according to relevance.

| Cluster | Number or Entries in Top 50 |
|---|---|
| Green | 31 |
| Red | 19 |

4.3.2. Two Clusters of Keywords Based on Co-Occurrences

Now that we understand the lay of the land in terms of the most frequent and relevant keywords, we will explore how these could be visualized in terms of bibliometric networks. Figure 3 showcases a network visualization of a term co-occurrence map, based on text data with weights based on occurrences. Figure 3 is related to Table 6 but now includes the full data set ($n_{term}$ = 720). The distance between two terms in the visualization approximately indicates the relatedness of the term in terms of co-occurrence. In general, the distance between two keywords demonstrates topic similarity and relative strength. The strongest co-occurrence links between terms are represented by lines. (This study also created maps with weights based on links and total link strength. However, these maps were virtually the same as the one below and did not show a significant shift. It was not possible to create a map based on the keywords' relevance.)

As shown in Figure 3, we found two very large, distinct, and dominant clusters. On the one side, the biggest cluster is the red one, consisting of 381 keywords. On the other side, the green cluster followed suit, with 332 keywords. Then, there are two almost indiscernible smaller clusters (mini clusters) in the middle space between these two fields. The blue mini cluster consisted of only 5 keywords, namely: "advertiser"; "gaming"; "gamer"; "music video"; and "commercial." Articles in this category for example discussed the quantitative gender representation in advertisements based on image analysis [25], computer and video games, and music videos. Then, there is also an almost indiscernible yellow mini cluster, which is comprised of solely two keywords: "engineering" and "STEM" (an acronym for science, technology, engineering, and mathematics).

When it comes to different types of media, Figure 3 shows that the clusters surrounding "film", "cinema", and "series" are very well connected. This means that film and television series are more extensively and effectively researched in the red cluster compared to, for example, the gaming and music industry. It is also interesting to note that the terms "feminist" and "feminism" find themselves at the complete outskirts of the red cluster. This implies that these terms are relatively marginal and predominantly co-occur with terms from the red cluster.

### 4.3.3. Temporal Shifts: Changing Research Agendas Indicated by Trends of Using Certain Keywords

We were also interested in seeing if one can discern a temporal shift when it comes to using certain terms in the discipline. Therefore, we made an overlay visualization. Its structure is identical to the network visualization, except that the keywords are colored differently to display their temporal shift, in terms of the years of their most frequent use in publications.

In Figure 4, the keywords are colored according to which year they were used the most. The default colors range from dark blue (earliest year), through turquoise and green, to yellow (most recent year). The color bar is shown in the bottom right corner of the visualization and indicates how years are mapped in colors. For example, terms colored dark blue were most frequently used 2010; terms colored turquoise were dominant in 2012, terms colored green showed up primarily in documents that were published in 2014; and terms colored yellow dominated publications in 2016. VOSviewer chose the period of 2010 until 2016 by looking at which years the respective terms peaked.

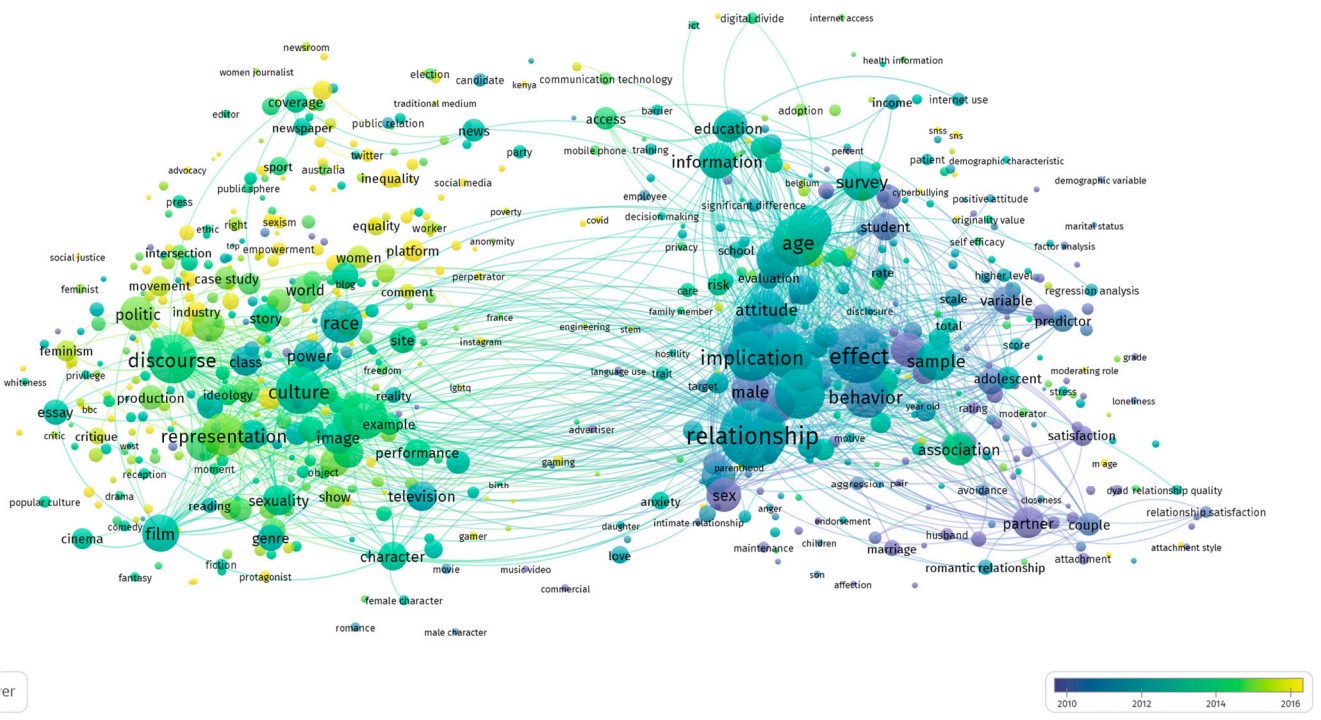

**Figure 4.** Overlay visualization of term co-occurrence map based on text data with weights based on occurrences.

Compared to the previous results of Figure 3, the new figure shows a clear temporal shift between the two dominant clusters. Terms belonging to the green cluster were most used in entries dated in 2010 and 2012, whereas terms belonging to the red cluster concerned entries released in 2014 and 2016. Furthermore, it is of interest to interpret Figure 4 in the context of the exponential growth of publications and citations mentioning the term "gender" in the Web of Science categories Film, Radio, Television or Communication (Figure 1). Even though the use of the term gender increased exponentially in communication, film, and media studies during the twenty-first century, its context has been changing. One can witness a clear shift between the green cluster to the red cluster.

## 5. Discussion

In this section, we share some reflections regarding the network analyses of the authors and text data. We also suggest some initial hypotheses for further research based

on our current results. Finally, we discuss the limitations of the corpus and the problem of grey literature.

### 5.1. The Network Visualizations of Authors and Keywords

The author map showed several interesting results. First of all, it is a novel and rather surprising finding that there exist seven distinct clusters of authors in the field of gendered communications, film, and media studies. However, the biggest clusters have several degrees of separation between them. From the red cluster, one needs to take several indirect steps before arriving at the green and dark-blue cluster, whereas the green and dark-blue cluster are better connected in terms of co-citations.

When it comes to the red cluster, several popular feminist film scholars find themselves at the outskirts of the cluster, as far as possible removed from the adjacent clusters. This means that they are mostly cited together with other authors in their cluster, but rarely ever in conjunction with authors from different clusters. For example, scholars such as Mulvey, McRobbie, and Gill find themselves on this periphery. This implies that these feminist film scholars are usually co-cited with people who approach gender from a theoretical and social constructivist viewpoint.

However, there are also feminist film scholars who do borrow from adjacent fields, such as Martha Lauzen and Stacy S. Smith. They are both categorized in the green cluster, and their nodes are located more closely to the adjacent yellow and dark-blue cluster. This means that they are more likely to be cited with authors from these adjacent fields. Lauzen and Smith's work can be categorized as using social-science-based research designs and methodologies in their gender-orientated film and media research. Their research shows how theoretical concepts can be backed up with "hard" data from the social sciences.

The visualizations also portray a tension in terms of the interdisciplinarity of the field. As mentioned in Section 4.1.4, many of the journals define themselves as cross-disciplinary. Both author and text clusters, however, indicate clear-cut boundaries between different scholarly traditions. There are several academics who act as main hubs between different clusters, such as Goffman, Hargittai, Bandura, and Eagly (see Figure 2). These main hubs facilitate links and connections to other clusters, which can be interpreted as an accurate representation of interdisciplinarity. However, the network map based on text data seems to present a more straightforward division between the humanities and social sciences, with the red and green clusters showing an unambiguous divide between the two paradigms.

Several clusters on the author map show differences when it comes to network density. For example, the red author cluster is very dense, with authors being so frequently co-cited together that they are almost stacked on top of each other (see Figure 2). Authors in the yellow cluster, on the other hand, are more loosely spread out over the map. The yellow authors are located in the middle of the map, and authors in clusters on the periphery often need to go through a yellow author in order to connect to a cluster on the opposite side of the map. This implies that even though authors in the red cluster are more likely to be co-cited together, authors in the yellow cluster (while belonging to a similar tradition as their fellow members) are more likely to be co-cited together with authors from other clusters. In the end, the yellow authors are the ones that bridge gaps between different traditions, and can, therefore, be considered significantly more interdisciplinary than authors in other clusters. As mentioned previously, the green cluster also holds a powerful position within the bibliometric network. The authors belonging to the green cluster possess robust ties to other authors in the network, whether they are within the green cluster or other clusters. To access the highly influential red or dark-blue clusters, authors must first pass through the green cluster, highlighting the interdisciplinary nature of the green cluster's authors, who serve as intermediaries between the various traditions in the field.

The keyword maps also display several interesting results. Firstly, they show that the term gender is used in two dichotomous ways. It is used either in relation to a more qualitative, or quantitative approach. The keyword maps seem to indicate that there are two very distinct academic traditions present in the field. On the one hand, gender is written

about in the context of "discourse", "culture", "class", "power", "race", "representation", and "image". On the other hand, it is used in a more formal social science setting, with terms including "relationship", "effect", "behavior", "sample", "survey", "age", "implication", "information", and "sex". Therefore, we can hypothesize that gender is used in either of two ways: the concept of gender as a social–cultural construct, or gender as a variable in more formal social and psychological quantitative research designs.

Similar to the feminist film scholars, the words "feminism" and "post-feminism" find themselves on the outskirts of the map. We expected these terms to be more in the center of the field, since feminism and post-feminism are two topics that are closely connected to the term gender. In addition, since the term feminism has been widely used for several decades, one would expect it to be more firmly situated in the field, and not on the periphery. This indicates that these conceptual terms are more likely to be used in a theoretical and social constructivist viewpoint, and not for methodologically and empirically oriented studies.

The existence of the two mini clusters (blue and yellow) in the keyword map (Figure 3) indicated that there are two traditions within gendered communications, film, and media studies that are rather small: research surrounding advertising, gaming, and music videos, and research focusing on engineering and STEM. The presence of these mini clusters can indicate two things. Firstly, research in these mini clusters is not fixed to one of the two dominant paradigms within the discipline. The yellow mini cluster is more closely located to the green cluster, whereas the blue mini cluster is nearer to the red cluster. Based on the text data, VOSviewer did not link these keywords to dominant clusters, but created two mini clusters instead. Another explanation for the small size of these mini clusters could be that their respective areas of research are still underdeveloped when it comes to addressing the topic of gender.

Thus, we hypothesize that the concept of gender could be used in two ways. In the red cluster, the term is used to refer to a social–cultural construct as a more theoretical approach. The green cluster, on the other hand, uses the term gender in more formal research settings, especially quantitative sociological or psychological research designs.

### 5.2. Initial Hypotheses Regarding the Clusters and Suggestions for Further Research

By identifying seven clusters of authors and two clusters of textual data (keywords), this research represents the first stage in exploring academic patterns in the field of gendered communications, media, and film studies. However, it is beyond the scope of this research to explore and interpret the internal characteristics of each of the seven author and two keyword clusters. Based on the conceptual framework and empirical results of the present study, it could be possible to conduct such a research project. A second study could identify and dive deeper into the characteristics—such as the exact academic orientations and internal structure—of all seven authors networks and the two textual networks.

With regard to the author clusters, each scholar´s academic output should be examined by looking at the most influential works in their respective oeuvre, and identifying the thematic, conceptual, and methodological similarities between them and other authors in their cluster. This cannot be completed by simplifying each author to a couple of keywords. Based on a detailed internal analysis of clusters, one could characterize the academic orientations of each cluster and give them proper, substantial names. It is beyond the scope of this research to examine all 995 authors ($n_{authors}$ = 995) in depth, but future research could explore this.

However, based on the present research, we may already articulate some observations. Among the seven authors' clusters (Figure 2), the red cluster can be seen as a multidisciplinary, theoretical cluster, involving cultural studies, sociology, and gender studies. The green cluster mainly focuses on media psychology and communications research and can be considered cross-disciplinary due to its positioning on the map. The dark-blue cluster mostly features psychology scholars and authors focusing on inter-personal communications. The yellow cluster predominantly concerns interdisciplinary scholars, with authors focusing on media and communications. Therefore, this cluster finds itself in the middle of

the map, between all the other clusters. The purple cluster involves the discourse analysists, scholars focusing on social roles, and linguistic analysts. The turquoise cluster focuses on news media, journalism, popular culture, and gender. Finally, the orange cluster showcases four Japanese scholars that look at gendered communications, film, and media studies.

Similar initial—and partially competing—hypotheses could be made regarding the textual clusters in Figure 2. Table 10 displays the three divides that could be hypothesized.

**Table 10.** Hypotheses on textual clusters.

| Red Cluster | Green Cluster |
| --- | --- |
| Mainly: | Mainly: |
| • Film, television, and radio studies | • Communication studies |
| • Humanities | • Social sciences |
| • Qualitative approaches | • Quantitative approaches |

These early hypotheses could be too simplistic as they may imply clear-cut divisions, even though this is not the case. Social sciences, of course, do have qualitative aspects, and many quantitative studies are carried out in film, television, and radio studies. Therefore, more in-depth research is needed to substantiate the character and the corresponding naming of these clusters.

### 5.3. Joint Considerations of the Author Map (Figure 2) and the Keyword Map (Figure 3)

The present study found that the structural patterns of the author map and the key-word maps are completely different. How is it possible that the same authors who clearly create seven different clusters in terms of co-citation, become dichotomous when it comes to the key terms of their research? There could be competing hypothetical explanations for this result. A possible explanation might be that each specific author cluster engages both qualitative and quantitative research approaches, and many specific authors could also adhere to this division. This would allow for the formation of diverse author groups (seven clusters) based on research themes, and a dichotomous pattern of keyword networks.

Another hypothetical explanation could be that one can divide the author clusters up into qualitative and quantitative parts. This means that the clusters that find themselves on the left side of the author map, can be identified with a more qualitative tradition. Author clusters that are located on the right side of the map follow a more quantitatively oriented methodology. The authors in the middle could be the ones that borrow from both qualitative and quantitative traditions. This could mean that the orange and red cluster follow more qualitative/theoretical traditions, whereas the dark-blue and green cluster represents the quantitative side. The purple, turquoise, and yellow clusters are more diverse in terms of qualitative–quantitative approaches (see Figure 5).

However, as previously stated, additional extensive research is needed to see if any of these hypotheses hold up. A deeper exploration and understanding are needed of both the author as well as the keyword clusters to see if this overlap is plausible. Moreover, the qualitative–quantitative divide is only one hypothesis concerning the dichotomy of the keyword maps. As discussed above, other competing hypotheses could be formulated that could account for the dichotomy in the keyword maps.

The truth could also lie somewhere in the middle between these possible hypotheses. One thing is certain, however. An author does not consciously or strategically place themselves in these respective clusters. They belong to these clusters by deciding who to quote, what theoretical traditions they follow, and what key terms they use in their work. Only further empirical research could validate the explanatory power of these hypotheses.

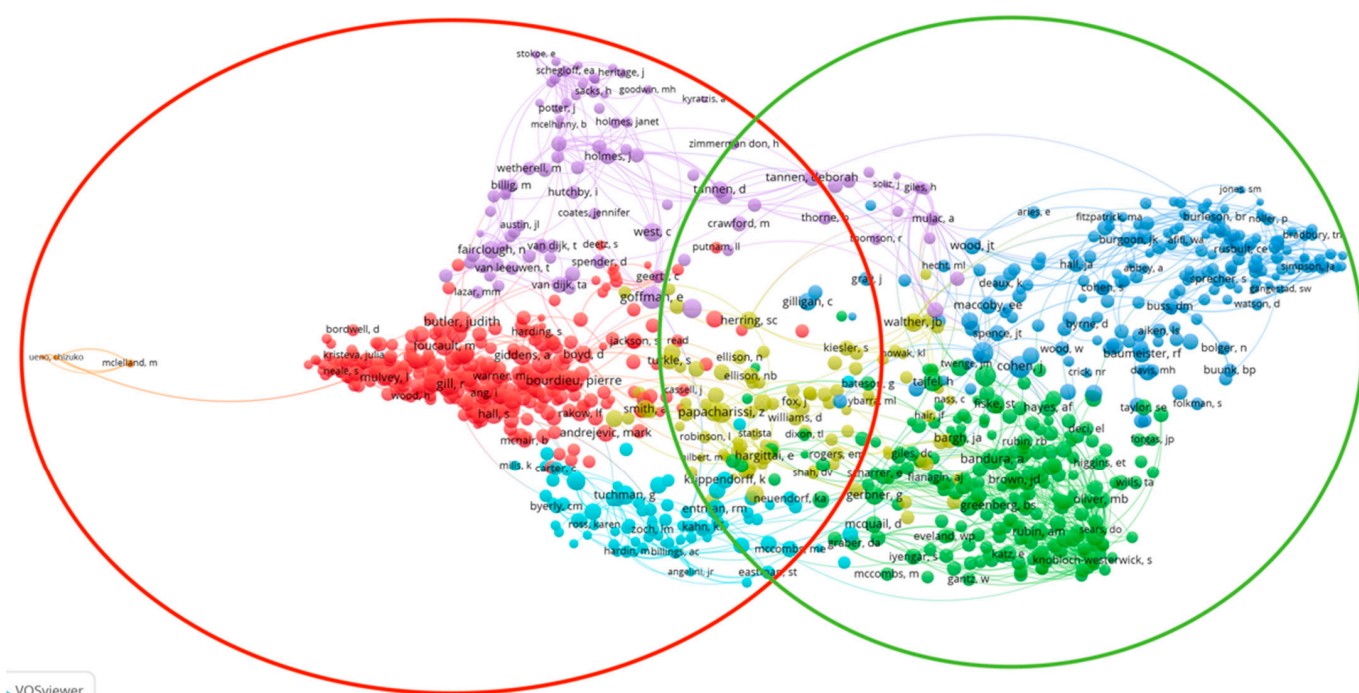

**Figure 5.** A combined visualization of the author map (Figure 2) and the keyword map (Figure 3).

*5.4. Limitations of the Corpus and the Problem of Grey Literature*

Being limited to the database of Web of Science, this study lacks a complete overview of all the existing research literature on gendered communications, film, and media studies. There are two limitations that are attached to using Web of Science. First of all, Web of Science could be characterized as a database that is more biased towards the science paradigm—as its name indicates—and, thus, relatively downplays the humanities and social sciences. Other databases such as Scopus or Elsevier, therefore, could provide a good alternative to or extension of the dataset. Additional research would be needed to include these databases.

Secondly, since Web of Science only includes peer-reviewed journals, scholarly books, and conference proceedings, it is natural that some of the literature is overlooked. This assumption was proven to be true when the author map only indicated small nodes for researchers such as Stacy S. Smith and Marta Lauzen. Throughout the years, both researchers have reported steadily on gender representation in the film and television landscape. Smith has not only examined on-screen portrayals of gender, race/ethnicity, LGBT and disability in 1300 popular films from 2007 to 2019 [26], but also off-screen representation when it comes to directors' gender and race/ethnicity across the same sample [27]. In a similar vein, Lauzen has examined portrayals of female characters in top grossing U.S. films [28], as well as looking at the behind-the-scenes employment of women in the same sample [29]. However, since most of their research is published on online platforms and the websites of their respective universities, these works are categorized as grey (non-academic or not strictly academic) literature and are not included in the Web of Science database.

Grey literature can be defined as publicly accessible, empirical, and policy research material that does not appear in strictly academic systems or channels of distribution, publication, or bibliographic control. It can be published at all levels of academia, but also appear in governmental, business, and industry publications [30]. The word "grey", therefore, refers to the uncertain status of this research literature, but not to its relevance nor ethical status. The uncertainty stems from the fact that grey literature is often difficult to access through mainstream academic databases, and its majority is not peer-reviewed [31]. However, prior studies that have noted the importance of including grey literature in

systematic literature reviews are Benzies, Premji, Hayden and Serrett [30], and Mahood, Van Eerd and Irvin [31]. They found that grey literature can be useful to validate the results of research-based academic literature searches [30], and it provides a more comprehensive view of the available evidence [31].

Significant grey literature studies exist that have conducted quantitative analyses of the representation of female characters in films. In fact, entire university research units are dedicated to studying the subject. Some influential examples include studies from the Hollywood Diversity Report team [32] and the Center for Scholars & Storytellers [33] at the University of California; materials from the Geena Davis Institute on Gender and Media Studies at the Mount Saint Mary's University [34]; the Annenberg Inclusion Initiative at the University of Southern California [35]; and the Center for the Study of Women in Television & Film at the San Diego State University [28].

Popular news outlets have actively covered these research reports, yet they are not included in the Web of Science Database. A possible implication is that the results of this study might give a somewhat skewed look of gendered communications, film, and media studies, since its sample does not include all available research publications in the field, most significantly leaving out grey literature.

## 6. Conclusions

This study set out to use bibliometric methods to explore and visually display network maps of the academic discipline of gendered communications, film, and media studies. To recall our original intentions and goals, this paper sought out to answer four research questions:

1.  Which authors contribute to gendered communications, film, and media studies?
2.  What clusters of authors exist in gendered communications, film, and media studies and how do these clusters relate to each other?
3.  What keywords/terms are most likely to be used in communications, film, and media studies in conjunction with gender?
4.  What keyword clusters exist in gendered communications, film, and media studies and how do these relate to each other?

These research questions were tailored into specifically designed data collection questions that could be answered by conducting bibliometric research in the Web of Science database. Then, two analyses were conducted: (a) a network analysis of the bibliographic data, specifically looking at "co-citations" (author map); and (b) a network analysis of the text data, specifically looking at the "co-occurrence" terms (keyword maps).

We found that the number of publications in gendered communications, media and film studies has been growing exponentially between the early 2000s and 2022. The top 50 authors in the field (1) in terms of co-citation are listed in Table 4. Gender theorists, gender sociologists, philosophers, psychologists, and social and cultural theorists (including Butler, Gill, Foucault, Bandura, Goffman, Hall, McRobbie, Eagly, Bourdieu, and Connell) lead the list of authors with the highest number of co-citations in the field. Some of these most co-cited authors tend to be generalists, and they are followed by specialists in the field.

We found that the 995 authors ($n_{authors}$ = 995) who publish in the field of gendered communications, film, and media studies (2) form seven author clusters, which we indicated with colors (see Figure 2). The size of the clusters differs significantly, with the authors being distributed as follows: red cluster (311), green cluster (211), dark-blue cluster (192), yellow cluster (99), purple cluster (95), turquoise cluster (83), and the orange cluster (4). We also looked at the extent to which other clusters cooperate with one another. The results suggest that this is different per cluster. For example, the red cluster—as one of the biggest clusters—is more secluded in terms of co-citations compared to the other two dominant clusters, the green and the dark-blue one. The density/looseness of the clusters also displays some variety. We can only hypothesize that the clusters could be approached as follows, but additional research is definitely needed to confirm this classification: (a) the red cluster features primarily theoreticians of gender and culture;

(b) the dark-blue psychologist cluster; (c) the green media psychology and communications cluster; (d) the purple sociology, linguistic and discourse analysis cluster; (e) the turquoise news media and journalism, popular culture, and gender cluster; (f) the yellow social media and communications cluster, and (g) the orange Japanese communications, film, and media cluster.

To discover which keywords/terms were most likely to be used in gendered communications, film, and media studies (3), this paper conducted a textual analysis of the same dataset. Tables 6 and 8 showcased the top 50 keywords according to number of occurrences and overall relevance.

The textual (keyword) networks in gendered communications, film, and media studies (4) form two main clusters and two mini clusters, as displayed in Figures 3 and 4. The visualizations feature an extensive scope of 720 keywords ($n_{terms}$ = 720). Our key finding is that as opposed to the seven clusters of authors, the textual clusters show distinctively different patterns. These form and are stratified along the lines of a dichotomous and bipolar divide. This means that although scholars in the discipline form seven author network clusters in terms of co-citations, their discursive communities display a divided, diverging, and bifurcated pattern.

The conceptual framework and empirical results of the present study lay the foundation for further research regarding the diverse academic agendas of the seven author clusters and interpret the split nature of the two keyword clusters as well as the key difference between the two patterns.

**Author Contributions:** Conceptualization, M.S. and K.B.P.; Writing—Original Draft Preparation, M.S. and K.B.P.; Writing—Second Draft: K.B.P.; Writing—Review and Editing, M.S. All authors have read and agreed to the published version of the manuscript.

**Funding:** This research does not receive external funding.

**Data Availability Statement:** Available upon request.

**Acknowledgments:** We would like to thank Georgiana Turculet and Alesia Zuccala for their initiative and support throughout the process. We would also like to thank the two anonymous referees as well as Alesia Zuccala and Márton Demeter for their valuable comments on earlier drafts of this paper.

**Conflicts of Interest:** The authors declare no conflict of interest. The funders had no role in the design of the study; in the collection, analyses, or interpretation of data; in the writing of the manuscript, or in the decision to publish the results.

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
