# Peer review of "Mapping Gendered Communications, Film, and Media Studies: Seven Author Clusters and Two Discursive Communities"

_publications, doi:10.3390/publications11010015_

Round 1

Reviewer 1 Report

Dear Authors,

This manuscript is of high interest to scholars studying gender.

Using a bibliometric analysis of co-citations of authors and textual occurrences of key terms, this manuscript shows how gender is used in publications in communication, film and media studies from 1975 to 2022. The results show which authors are most influential, how these authors are connected, what keywords are predominant, and how these are connected.

First of all, the manuscript is well-written and the review is well-conducted. Secondly, the manuscript is truly enriching for researchers interested in gender, since it maps the most important scholars in the field as well as the main keywords in the area and also shows how these are connected. This is very valuable for researchers in the field. Finally, I am not aware of any similar research in the field, which is another major strength.

My only suggestion is to re-read the manuscript for fixing a few typos.

Best of luck for your revision!

Author Response

Dear sir/madam, 

First of all, thank you for taking the time to review our article. 
We greatly appreciate your feedback, and we will incorporate it to improve our manuscript. 
Again, thank you for your help!

All the best, 

Kim and Miklos

Reviewer 2 Report

It was a pleasure to read this manuscript, as its topic of study was very interesting and much-needed. Overall, I think the study will make for a valuable contribution to this journal. However, several items need to be addressed before getting it to a publishable state:

1.) The term "communication" was used in the context of the term "communications" at times. This occurred throughout the first half of the manuscript. Toward the latter half of the manuscript, the term "communication" seemed more appropriately used. As such, it would be beneficial to clarify the differences between the terms. For instance, "communication" is very specific to the areas of rhetoric and persuasion, while "communications" is more in the vein of film and media studies. Given this nuance in meaning, it could have impacted your findings here. 

2.) The term "field[s]" is overused in this manuscript to the point that it seemed like better terms could have been utilized in many cases. For instance, the terms "discipline" and "paradigms" would be more appropriate at times. For instance, paradigms might be better suited when discussing qualitative and quantitative research paradigms.

3.) Why did this study start at 1975, not 1972, thereby going for a 50-year span of analysis? 

4.) The writing could be tightened up in a lot of areas, as it seemed overly wordy throughout. Similarly, there were extra words that needed to be omitted altogether in many sentences throughout the manuscript. 

On a final note, I was genuinely surprised to not see certain journals listed like Women's Studies in Communication and Television & New Media. Both have been around for a long time and consistently publish articles that focus on gendered film and television research. 

Author Response

Dear sir/madam,

First of all, thank you for taking the time to review our article.

We greatly appreciate your feedback, and we will incorporate it to improve our manuscript.

Below, you will find our answers to your comments.

1.) The use of the term “communication(s)”: thank you for highlighting this discrepancy. We have gone over the text by highlighting all the instances in which we use the term, and we have adapted them accordingly.

2.) The use of the term “fields”: thank you for this suggestion. We have gone over our paper and replaced the use of the word “fields” with “paradigms” where necessary.

3.) The start of the study: the year 1975 was chosen as a start date because that is the first time the database noted that there was a “gender” entry in the field of communications and films and media studies. The Web of Science database, thus, indicated that before 1975, no journals in their database were discussing the term “gender” in our designated fields.

4.) Writing: our other peer reviewer made the same comment, so thank you for bringing this to our attention. We will go over our manuscript again with extra detail in regards to its grammar, spelling and sentence structure.

5.) On behalf of your final note: it is possible that these journals are not listed or recorded in the Web of Science database, but, rather, in a competing database such as Scopus.

Again, thank you for your help!

All the best,

Kim and Miklos